# Stepwise Discovery of Insulin Effects on Amino Acid and Protein Metabolism

**DOI:** 10.3390/nu16010119

**Published:** 2023-12-29

**Authors:** Paolo Tessari

**Affiliations:** Department of Medicine, University of Padova, 35128 Padova, Italy; paolo.tessari@unipd.it

**Keywords:** amino acids, leucine, anabolism, catabolism, intracellular signals, isotopes, lean body mass, protein degradation, protein synthesis, turnover

## Abstract

A clear effect of insulin deficiency and replacement on body/muscle mass was a landmark observation at the start of the insulin age. Since then, an enormous body of investigations has been produced on the pathophysiology of diabetes mellitus from a hormonal/metabolic point of view. Among them, the study of the effects of insulin on body growth and protein accretion occupies a central place and shows a stepwise, continuous, logical, and creative development. Using a metaphor, insulin may be viewed as a director orchestrating the music (i.e., the metabolic effects) played by the amino acids and proteins. As a hormone, insulin obviously does not provide either energy or substrates by itself. Rather, it tells cells how to produce and utilize them. Although the amino acids can be released and taken up by cells independently of insulin, the latter can powerfully modulate these movements. Insulin regulates (inhibits) protein degradation and, in some instances, stimulates protein synthesis. This review aims to provide a synthetic and historical view of the key steps taken from the discovery of insulin as an “anabolic hormone”, to the in-depth analysis of its effects on amino acid metabolism and protein accretions, as well as of its interaction with nutrients.

## 1. Introduction

Since the discovery of insulin and the start of successful treatment of the diabetic condition by means of pancreatic extract(s) and later with insulin [1,2], it was immediately clear that the physiological and metabolic effects of insulin, beyond those on glucose and ketones, were “anabolic” in respect to the nitrogen-containing substrates, i.e., amino acids and proteins. Insulin promotes the maintenance and recovery of lean body and skeletal muscle mass (the latter composed of proteins by ≥40%) and guarantees a physiological body growth in infancy and adolescence. As biochemical analyses became increasingly available, previously unknown effects of either insulin deficiency on the elevation of amino acid plasma concentrations and urinary urea excretion, i.e., on nitrogen-containing substrates, or, conversely, of their reduction following insulin administration, could be demonstrated. Therefore, interest in the role of insulin in the regulation of amino acid and protein metabolism has progressively and markedly increased, paralleling the availability of novel analytical methods. 

The timing of the technological advancements allowing a deeper insight into the physiological mechanism(s) through which insulin regulates body nitrogen metabolism, composition, and growth, as well as muscle mass, are schematically reported on Table 1. The sequence of the investigational methods has moved from the clinical evaluation of “diabetic” subjects and from observations on experimental diabetes in animal models [1,2], to basic biochemical measurements (urinary excretion of non-protein nitrogen compounds [3] and plasma amino acid concentrations). Since then, these early observations have been extended to in vivo studies of amino acid metabolism, focused on end-point metabolites (non-protein nitrogen substances like urea, 3-methylhistidine, etc.) at first, then enriched by the use of a kinetic approach in whole-body turnover studies, based on isotope-dilution methods (using radioactive and/or stable isotopes of tracer amino acids). These techniques have been implemented by the arterial–venous catheterization of accessible organs, as well as by measurements of the incorporation of tracer amino acid isotopes either into tissues (skeletal muscle, heart, gut, etc.) through biopsy or into circulating proteins. Very recently, advanced isotopic/imaging techniques have been developed. Furthermore, the molecular mechanisms of the anabolic effect of insulin have been investigated by measuring intracellular mediators of insulin signaling, sometimes in conjunction with in vivo kinetics studies. Such an impressive battery of analytical techniques has expanded our knowledge of the mechanism(s) of insulin, as well as of insulin deficiency, on nitrogen-containing compounds.

## 2. Early Studies on Human Amino Acid Metabolism

Seminal metabolic studies were performed by Felig et al. in the second half of the last century. The major biochemical abnormalities in nitrogen-containing compounds observed by these investigators were those on branched-chain amino acids (BCAAs). Their concentrations were found to be markedly elevated in insulin-deficient diabetes, particularly in the extreme condition of ketoacidosis [4]. A two- to three-fold increase was reported in the concentrations of plasma leucine, isoleucine, and valine, and of α-amino butyrate, later and correctly attributed to increased release of BCAAs from endogenous protein degradation, and associated with increased urea excretion. Due to its osmolality, urea contributed, along with glycosuria, to the dramatic water loss of ketoacidosis [5]. The amount of urinary nitrogen excretion could be correlated with glycosuria [6,7], suggesting a common pathogenic factor for these two alterations. In contrast, glycoproteins, as well as some acute phase proteins, were found to be increased [8]. 

In diabetic ketoacidosis, changes opposite to those on BCAAs were described for other plasma amino acids. A ≈ 25–40% reduction was observed for the concentration of key glycogenic amino acids such as alanine, glycine, threonine, and serine, compatible with an increased glucose production from gluconeogenesis [4], later confirmed with more sophisticated techniques [9]. These abnormalities were proposed to contribute to the development of acute hyperglycemia and were reduced and/or abolished by insulin administration [10]. Another indicator of muscle-protein degradation, urinary 3-methylhistidine excretion, was found to be increased (by ≥40%) in diabetic subjects, even when receiving their usual insulin dose, suggesting a (partial) failure of insulin (presumably administered in suboptimal doses and/or patterns at those times) to fully normalize protein catabolism [11]. A sharp, dose-dependent decrease of plasma [12,13] as well as intracellular amino acid concentrations [14], particularly of the BCAAs, following insulin administration, was constantly observed. In addition, insulin stimulated amino acid transport [15], particularly A and X–C systems [16], in in vitro studies. Notably, insulin and amino acid levels were found to be strictly connected, not only because of the effect of insulin on the amino acids but also because the latter directly regulates insulin secretion in the presence of glucose [16,17].

Although not wishing to obscure the insulin role, an important observation was that of a “mass effect” of plasma amino acid concentrations, on their own uptake by tissues. Both splanchnic and peripheral (i.e., skeletal muscle) amino acid uptake, following either intravenous or oral amino acid load administrations, was shown to be (partly) independent of insulin [18]. 

## 3. The Effects of Insulin on Organ Amino Acid Exchange

Moving from plasma concentration data, a further step forward was represented by in vivo studies looking at the sites(s) in the body involved in the regulation of amino acid metabolism. Seminal organ catheterization studies were again performed by Felig et al., who investigated the amino acid net release into, or net uptake from, the circulation, both across the leg (i.e., a peripheral tissue) and the splanchnic area, in both the healthy subjects and in the insulin-deficient diabetic condition [19]. In diabetic patients withdrawn from insulin for 24 h, no consistent either net uptake or net output across the leg, of the branched chain amino acids was observed, as opposed to negative arterial-venous differences, indicating a net release, observed in the control group. Conversely, consisten, positive arterial-hepatic venous differences (A-HV) for alanine, serine, glycine, threonine, tyrosine, cystine, methionine, phenylalanine, and taurine, were observed, indicating a net uptake by the splanchnic area of these amino acids, in both the insulin-withdrawn diabetic and the control subjects. In both groups, the alanine fractional uptake exceeded that of all other amino acids, and was 1-fold greater in the diabetic (66%) than in the control subjects (36%). Conversely, the elevate arterial valine, leucine and isoleucine concentrations of the diabetic subjects could not be accounted for by an altered either splanchnic or peripheral exchange of these amino acids [19]. The apparently-negative results about the amino acids, of these otherwise seminal, catheterization studies, could be attributed to the large inter-subject variability of the data, as well as to the absence (at that time) of the isotope dilution approach that could have casted a deeper light into the underlying pathophysiological mechanism(s).

## 4. Whole-Body Amino Acid and Protein Kinetic Studies

Starting from the last decades of the last century, the study of the pathophysiology of amino acid metabolism in vivo was markedly enriched using isotope-dilution techniques, allowing the performance of kinetic studies, which were also applied to the evaluation of the insulin effects on amino acid turnover in vivo. The essential amino acid leucine was one of the most largely employed tracers for several reasons: (1) Leucine is an essential amino acid. Therefore, its release into the circulation, as well as its net catabolism (through irreversible loss via oxidation), closely reflects that of protein; (2) leucine is relatively abundant in body tissues and compartments. Therefore, it could be measured with confidence using appropriate methodologies; (3) leucine concentration is very sensitive to insulin levels, thus representing an ideal tracer to test the insulin effects; (4) leucine was found to exert a specific anabolic and biochemical effect, thus raising the interest on its metabolic role in vivo. Nevertheless, other essential amino acids, such as phenylalanine [20], methionine [21,22], and valine [23], have been used as suitable protein turnover tracers.

The use of turnover techniques also allowed the dissection of the in vivo effects of insulin between the two opposite processes of protein synthesis and degradation, therefore casting a new light on the mechanism(s) determining net protein anabolism. Although insulin had an undisputed anabolic effect on tissue protein net accumulation, the mechanism(s) of such an effect, i.e., a decreased endogenous proteolysis vs. an increased protein synthesis, had not yet been investigated in vivo. Protein degradation (i.e., proteolysis) is traced by the rate of appearance into the circulation of an essential amino acid that, under post-absorptive conditions, derives only from endogenous protein breakdown. Under feeding conditions however, the total rate of amino acid appearance into the circulation derives from the combined effect of (endogenous) proteolysis and of dietary protein absorption, which therefore needs to be estimated separately (see below). On the other hand, protein synthesis, i.e., amino acid utilization for protein assembly, can be indirectly traced by subtracting, from the total disposal rate of the tracer essential amino acid, that of its irreversible catabolism (i.e., oxidation for leucine, valine, and methionine, or hydroxylation for phenylalanine). Subtle but important details in such measurements are the choice of the “precursor pool(s)” used in the kinetic calculations. The “precursor pool” commonly refers to the isotopic labeling of the chosen tracer amino acid at the precise site, i.e., in the intracellular compartment(s), where the metabolic process (proteolysis, protein synthesis, amino acid catabolism, etc.) occur. The precise measurement in vivo of the “precursor pool” in accessible compartments is difficult. Although the easiest, most accessible site(s) where to measure the “precursor pool” in vivo would obviously be circulating blood or plasma, values in these compartment(s) commonly depart to some extent from the “real” intracellular ones. The latter could be better determined by tissue biopsy, also combined with subcellular compartment analyses, as well as approached using “surrogates”. Examples of surrogates are the deamination products of either leucine (i.e., plasma α–isocaproic acid (KIC)), or valine (ketovaleric acid (KIV)), both measurable in plasma, intracellular aminoacyl-tRNA (measured by biopsy), the EAA labeling attained into a fast-turning-over protein measured at the plateau value, mathematical modeling of measured variables, etc. A model based on arterial–venous measurements, combined with tissue (skeletal muscle) biopsy, has been proposed too (see below). As a specialized investigator can appreciate, the result(s) offered by each of these methods need to be balanced against the analytical precision, the in vivo feasibility and invasivity, the complexity of the experimental design(s) and of data calculation and interpretation, the available technological facilities, and their costs, all aspects that cannot be discussed in this review. 

A key variable, however, needs to be considered and handled in studies focused on the effects of insulin administration in vivo. Indeed, the effects of a systemic insulin administration on protein degradation and synthesis may be hampered by the concurrent decrease of plasma amino acid concentrations and, conversely, by their increase when insulin is withdrawn. In other words, the insulin-induced changes in amino acid concentrations (i.e., modifying their “mass-effect”) would interfere with the evaluation of the insulin effect itself. Such a variable can be controlled and/or circumvented in one way, by employing the so-called “amino acid clamp” technique during systemic insulin administration. Such a technique, similar to that employed in studies of glucose metabolism, consists of the maintenance of the basal plasma amino acid concentrations by calibrated exogenous amino acid infusions during the insulin infusion/administration. Alternatively, another way to control for such a variable is to infuse insulin directly into the artery perfusing a given organ (mostly the leg or the forearm, representative of skeletal muscle), thus avoiding a systemic (i.e., arterial) hyperinsulinemia and the concurrent suppression of the plasma amino acid concentrations. By any of these approaches, the “isolated” effect(s) of insulin, either at the whole-body level or in a perfused limb or organ, can be selectively investigated.

When insulin is infused systemically without prevention of hypo aminoacidemia, whole-body protein degradation was universally found to be suppressed, in a dose-dependent fashion [12,13,24]. The semi-maximal effect of insulin in suppressing whole-body protein degradation occurred approximately between 20 and 30 μU/mL of plasma insulin concentration, whereas the “maximal” effect was obtained by pharmacologic hyperinsulinemia (1000–1500 μU/mL) [12,13]. The insulin-induced reduction of plasma amino acid concentration and release from endogenous proteolysis could be demonstrated in type 1 diabetes subjects too [25,26,27,28,29,30], also in the post-prandial state [31]. Thus, the decreased concentrations of leucine, as well as of other EAAs, following systemic hyperinsulinemia and, conversely their increase in the insulin-deficient condition of type-1 diabetes, could be positively related to concurrent changes in endogenous protein breakdown. Another index of the insulin-induced suppression of protein catabolism was the reduction of the urea-nitrogen rate of appearance (Ra) in burned and septic patients treated with insulin [32]. 

Conversely, at variance with the inhibitory effect on endogenous proteolysis, the (expected) effect of insulin on the stimulation of whole-body protein synthesis (as indirectly determined by the irreversible loss of an essential amino acid (mostly leucine)) was not reported by any of the investigators [12,13,28,29,33,34,35,36,37]. Surprisingly, whole-body protein synthesis was either unchanged or even decreased following systemic hyperinsulinemia. Such a paradox has puzzled investigators since the first reports on this issue and forced them to design a variety of experimental designs aimed at resolving this otherwise apparently physiological observation. Briefly, the failure to observe a stimulation of whole-body protein synthesis by insulin was predominantly attributed to the concurrent insulin-induced hypoaminoacidemia, as discussed above. Nevertheless, in some studies where plasma leucine and other amino acid concentrations were clamped at baseline, either physiologic or even pharmacologic hyperinsulinemia failed to stimulate protein synthesis [38,39]. In contrast, a marked stimulation of protein synthesis occurred when plasma amino acid (including leucine) concentrations were raised, independently from the increase of insulin [29,35,40,41]. Moreover, under controlled conditions of comparable plasma amino acid concentrations at either low or high insulin concentrations, the amino acid-induced stimulation of whole-body protein synthesis (in comparison to basal rates) was not augmented further by hyperinsulinemia [40]. 

The specific focus of this Special Issue does not allow to get into deeper details about this complex issue. From a physiological, “finalist” standpoint, it could be reasonable to attribute to insulin a predominant role in the suppression of endogenous protein breakdown, through a hormonal, intracellularly-delivered signal that does not require the provision of either metabolic precursor(s) or energy substrates. The study of the mechanism(s) of the suppression of protein breakdown, as well as its role and significance in the removal of aged or modified proteins, is at the core of current research. Conversely, at variance with the suppression of proteolysis, the stimulation of protein synthesis would require both the provision of precursor substrates, i.e., the amino acids, and of hormonal signals, therefore explaining, at least in part, the above-mentioned “paradox”. Notably, such a combined effect of insulin and amino acids physiologically occurs after a mixed meal in healthy subjects.

## 5. Organ and Tissue Protein Kinetic Studies

The comprehensive measurement of amino acid kinetics in accessible, selected districts or organs, such as a limb (i.e., the forearm or the leg, representing skeletal muscle), the splanchnic area, the heart or the kidney, including rates of protein synthesis and degradation, became available by combining tracer amino acid infusion with the organ arterial–venous catheterization. Such an approach requires sampling in an artery and in the deep-vein draining the organ, the measurement of organ blood flow [27,33], and, in some instances, also of the amino acid tracer in the tissue by biopsy [25,36,37,41]. All these parameters have also been combined in an unifying model [42]. Clearly, these techniques are complex, somehow invasive, and require a sound analytical apparatus. The results from a variety of studies using such techniques could be summarized as follows.

By combining organ arterial–venous catheterization and isotope-dilution techniques, using either the forearm or the leg model, a systemic insulin infusion, besides reducing significantly arterial amino acid concentration [24,27], abolished the net amino acid release (i.e., improved amino acid balance) by human muscle in the post-absorptive state [43,44,45], by inhibiting protein breakdown with no effects on protein synthesis [43,45], and strongly augmented the positive limb amino acid balance already seen in experiments where only amino acids were infused [41]. Regarding the response to insulin of individual amino acids, the net leg balance of tyrosine (a non-essential amino acid) was not affected by hyperinsulinemia at all, whereas that of BCAAs and methionine was switched from net release towards net uptake [46]. However, leg arterial 3-MH concentration and exchange (a marker of muscle proteolysis) were not significantly affected by 2 h of hyperinsulinemia [46]. Taken together, these data indicate that the anabolic effect of systemic hyperinsulinemia at the skeletal muscle level are due to inhibition of protein breakdown. 

However, when the fall of plasma amino acid concentrations was prevented, either by means of the amino acid clamp or following the intra-arterial insulin infusion, the results were somewhat contrasting. In some studies, insulin did not increase muscle-protein synthesis, whereas it inhibited muscle-protein degradation [38,43]. In contrast, in subsequent studies applying a complex compartmental model across the leg, the intra-arterial insulin infusion was found to increase muscle-protein synthesis, whereas it did not affect muscle-protein degradation [42,47]. Notably, in T1DM patients the infusion of insulin did not stimulate muscle-protein synthesis even when combined with hyperaminoacidemia, suggesting a reduced response compared with that of non-diabetic subjects [25,37].

Notably, the effects of insulin on muscle metabolism could also depend, at least in part, on changes in blood flow. Insulin infusion, either in the whole body or directly into a catheterized limb (leg or forearm), generally stimulates blood flow [42,43,47]. Physiologic, systemic hyperinsulinemia stimulates whole-body nitric oxide production, a key regulator of blood flow [48]. Therefore, simply by such an insulin-driven stimulation, the increase delivery of anabolic substrates (mainly the amino acids but also the energy-precursor glucose) to skeletal muscle could sustain protein synthesis. 

The effects of insulin administration of amino acid kinetics across the splanchnic bed were also investigated in vivo. In post-absorptive healthy volunteers, the net splanchnic amino acid uptake was positive, an effect due to rates of protein synthesis exceeding protein breakdown [45]. Following a systemic insulin infusion, the net positive splanchnic net balance decreased due to the reduction of protein synthesis without changes in protein breakdown [45]. Thus, the mechanisms of the insulin’s anabolic effect appear to be different between peripheral tissues and the splanchnic area. Insulin was anabolic in muscle mainly by inhibiting protein breakdown, by enhancing the AA-induced protein synthesis, and by decreasing leucine transamination [45]. Conversely, in the splanchnic region, insulin decreased protein synthesis and net uptake of the amino acids with no effects on protein breakdown [45]. The net positive protein balance in the splanchnic area is thus sustained by amino acid release from peripheral tissues, highlighting a key muscle-splanchnic relationship. 

It should also be considered that the different responses of tissue protein turnover to insulin may also be time-dependent, due to the different fractional turnover rates of tissues. Fractional protein turnover is low in skeletal muscle proteins (1.2–2.4% per day^−1^) and greater in the liver (≈23%), kidney (≈20% or gut proteins (≈8–30%) [49]. The acute insulin effects would likely be more easily detected in fast-turning-over tissues.

Another organ studied with the catheterization approach is the kidney. In diabetic patients withdrawn from insulin treatment for 24 h, renal amino acid exchange was similar to that of healthy individuals, suggesting also that the kidney was not an important gluconeogenic organ in human diabetes [50]. Later studies, however, based on techniques not yet available previously, suggested that, although quantitatively unimportant, the kidney may be a gluconeogenic organ in humans [51,52]. With respect to the metabolism of other amino acids, insulin deprivation was associated with a net renal uptake of phenylalanine and a net release of tyrosine derived from renal phenylalanine hydroxylation. However, the global amino acid metabolism across the kidney was not altered in either insulin-deficient diabetes or insulin deprivation [50,53].

Finally, modern approaches combining amino acid isotope infusion and imaging techniques, such as positron-emission tomography, may cast new light on the effect of insulin on amino acid metabolism in otherwise inaccessible organs such as the brain [54].

## 6. Extending in Vivo Kinetic Studies to Molecular Mechanism(s) of the Anabolic Action of Insulin

The role of intracellular mediators on the effects of insulin on amino acid and protein metabolism in tissues and isolated cells has become another key focus of interest, and has been increasingly studied, starting in the last decades of the 1990s. Both in vitro [55] and in vivo [47] experimental designs and models have been employed.

Muscle-protein degradation is under the control of two major mechanisms: the ubiquitin–proteasome and the autophagy–lysosome systems [56,57]. These systems are activated in muscle atrophy and variably determine the loss of muscle mass. They are dependent on transcription program(s) controlling the expression of rate-limiting enzymes. Insulin deeply affects these systems by means of a complex signaling network rather than by a simple, linear cascade, further modulated by signaling molecules downstream of the insulin receptor, involving many receptor tyrosine kinases (RTKs) [58]. Both the insulin and IGF-1 receptors are involved in the maintenance of muscle mass and the regulation of proteostasis [59,60,61].

Classically, the biological effects of insulin are initiated by hormone binding to its receptor, a member of a large family of receptor tyrosine kinases (RTKs), followed by the activation of class I phosphoinositide 3-kinase (PI3K), which in turn activates Akt and the target of the rapamycin (mTOR) complex 1 (mTORC1) network, which ultimately modulates cellular metabolism and growth [58]. Central in this pathway is the activation of mTORC1 [60,61], a complex molecule composed of mTOR, Raptor (i.e., a regulatory-associated protein of mTOR), and other proteins. Activation of mTOR is associated with signals coming from amino acids, glucose, oxygen, energy (ATP), and growth factors (hormones such as insulin, IGF-1, and cytokines). The activation of mTOR though IRS/PI3K/Akt by either insulin or IGF1 controls common downstream molecular pathways that include the MAPK pathways, the phosphorylation of p70 ribosomal protein S6 kinase 1 (S6K1) [62] and the eukaryotic initiation factor 4E-binding protein 1 (4E-BP1) [63]. 

Akt activation conveys many of the physiologic effects of insulin (as well as of IGF-1) in muscle, such as enhanced glucose uptake, muscle growth, protein synthesis, and degradation. Knowledge of insulin regulation of proteostasis [61] is essential to understanding the insulin effects both in normal physiology and in pathological conditions and/or mechanisms arising from insulin-deficient conditions, such as type 1 diabetes. 

Another group of transcription factors, the forkhead box O (FoxO) family, is associated with the effects of insulin in tissues and cells. The FoxO family includes four isoforms, that mediate the action of either insulin or insulin-like growth factors, on cell growth, differentiation, oxidative stress, senescence, autophagy, and aging [64,65]. In muscle, FoxO transcription factors are associated with the control of muscle-protein degradation and autophagy, and in the liver, they sustain gluconeogenesis and glycogenolysis [66]. Insulin, through the Akt pathway, negatively regulates the transcription factor FoxO, one major effector of muscle atrophy, thus inhibiting proteolysis [67]. Signaling mediated by insulin receptor (IR) binding appears to be more important than IGF-1 signaling in controlling gene expression in differentiated muscle [60]. Thus, the anabolic effect of insulin on muscle-protein anabolism, in addition to increased protein synthesis, is due also to the reduction of protein breakdown through FoxO inhibition [65,67]. 

## 7. Conclusions

This brief review shows that, since the discovery of insulin about one century ago as a major glucose-regulating hormone and an essential treatment of type 1 diabetes mellitus, major advances have been gained in the study of the effects of insulin on protein metabolism at the whole-body level, as well as in organs and tissues and at molecular levels. New techniques will likely help to obtain a deeper insight into the effects of this life-saving hormone on amino acid and protein metabolism. 

## Figures and Tables

**Table 1 nutrients-16-00119-t001:** The historical appearance of major investigational steps in the study of the pathophysiology of amino acid and protein metabolism in type 1 diabetes.

Parameter	Methodology	Year
“Melting down of the flesh into urine”	Aretaeus description	≈200 a.C.
Blood urea concentration and urine urea excretion in experimental diabetes	Measurements in the diabetic condition and after correction by a pancreatic extract	1916
Electrolyte balances *w/o* and with insulin	Analytical methods	1933
Plasma amino acid pattern and concentration	Plasma measurements	1970
In vivo organ balance catheterization studies	Organ arterial–venous catheterization	1971
Insulin signaling on tissue protein turnover	Intracellular mediators	1982
Whole-body proteolysis and protein synthesis	Amino acid turnover studies	1984
Handling of mixed meals by dual isotope techniques	Intravenous and oral amino acid isotope administration	1988
Muscle-protein synthesis	Isotope infusion, arterial–venous leg catheterization, and muscle biopsy	1990

## Data Availability

The data reported in this review article are freely accessible in the scientific literature.

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
