# Peer review of "Stepwise Discovery of Insulin Effects on Amino Acid and Protein Metabolism"

_nutrients, 2023, doi:10.3390/nu16010119_

Round 1

Reviewer 1 Report

Comments and Suggestions for Authors

Summary: The aim of this manuscript was to write a review and synthesis of evidence since the discovery of insulin as a glucose regulating hormone and insulins effects on amino acid and protein metabolism. The main contribution of this manuscript is that it adds to the literature by providing a synthesized overview of the effects of insulin on amino acid and protein metabolism in a historical stepwise manner. Strengths of this study are that it is well written and is an interesting read documenting the continued evolution of the body of knowledge on this topic from discovery to present day in a logical flow.

General comments: Overall, this review is well structured and written, and is an interesting read. This manuscript adds to the literature by providing a synthesized overview of the effects of insulin on amino acid and protein metabolism in a historical stepwise manner documenting the continued evolution of the body of knowledge on this topic from discovery to present day. I have no concerns or issues with flaws of this manuscript. However, there are a number of minor grammatical errors in the manuscript that need to be rectified with most occurring in the latter text of the manuscript.

Specific comments:

Line 18: Regarding the part of the question, “why do we eat”, the content of this manuscript focuses on the stepwise discovery of insulins effects on amino acid and protein metabolism but the content does not address the question of “why do we eat” which is broad question encompassing factors including social, cultural, behavioral, neurobiology, food environment, and economic among others. I would suggest excluding this question and potentially the last sentence or framing this sentence and question in a more aligned way with the content of the manuscript.

Comments on the Quality of English Language

Minor corrections of grammar mistakes are required.

Author Response

I thank the Reviewer for her/his useful comments. I removed the sentence: "...why do we eat”. I agree that it is somehow out of the main flow and the focus of this review. I tried to amend the gramatical errors. I added a brief paragraph expanding the role of insuoin actino al molecular level.

Reviewer 2 Report

Comments and Suggestions for Authors

This article summarizes foundational studies on the role of insulin in protein turnover. This is an important topic where a compilation of older studies can provide important insight. The article could be a bit more analytic in style because of hindsight available today. The last paragraphs treat the topic quite superficially and could benefit from a more systematic approach.

The article needs language editing particularly the use of prepositions. 

line 22, grammar: if you use "with" with insulin also use "with" with pancreatic extracts.

line 26, grammar: composed of 40% protein.

line 29/30: please indicate direction of the observed change.

Line 35-44: please break down this convoluted long sentence into several more coherent sentences. Check the grammar of each sentence.

Line 59-60: "Correctly attributed to increased release of BCAA". This is not quite correct. Plasma AA levels are mainly set by metabolism not by protein breakdown. (Holeček, Milan. "Why are branched-chain amino acids increased in starvation and diabetes?." Nutrients 12.10 (2020): 3087.) Phillip J. White, Robert W. McGarrah, Mark A. Herman, James R. Bain, Svati H. Shah, Christopher B. Newgard, Insulin action, type 2 diabetes, and branched-chain amino acids: A two-way street, Molecular Metabolism,

Line 60: associated with

Line 65: described for

Line 66: observed for

Line 73: failure of

I stopped correcting grammar from here because errors are too frequent.

Line 78: The stimulation of amino acid uptake by insulin in muscle goes in hand with the anabolic effect on this tissue. Kashiwagi, Hitoshi, et al. "Regulatory mechanisms of SNAT2, an amino acid transporter, in L6 rat skeletal muscle cells by insulin, osmotic shock and amino acid deprivation." Amino Acids 36 (2009): 219-230.

 Line 84/85: amino acid uptake...was partly independent of insulin.

Line 144: Please define unit and the volume here.

Line 149: Please see comments above for line 59/60.

Line 151: Please explain "Ra"

Line 165: "However" might not be the right conjunction here.

Line 186: Your summary suggests that Insulin mainly regulates protein breakdown and less so protein biosynthesis. This might be worth expanding or clarifying. Mechanistically this is interesting because of the different response of the splanchnic bed.

Line 189: "Very naively, such a condition is that physiologically occurring following either protein or mixed oral nutrition, that stimulate insulin secretion too." Please rephrase: Such a condition most likely occurs physiologically after a mixed meal in the presence of insulin.

Line 207/208: "net" used twice.

Line 219: Thus, it is likely

Line 230: How about glutamine metabolism. As the main precursor for glucose this might be downregulated by insulin.

Comments on the Quality of English Language

The article needs extensive editing. I have made some suggestions but due to the quantity of errors I gave up listing all errors. 

Round 2

Reviewer 2 Report

Comments and Suggestions for Authors

The author has addressed most of the comments made by the reviewer.

Comments on the Quality of English Language

The use of English is acceptable in the revised version.